# Contribution of Non-Timber Forest Product Valorisation to the Livelihood Assets of Local People in the Northern Periphery of the Dja Faunal Reserve, East Cameroon

**Manfred Aimé Epanda [1,*], Romaric Tsafack Donkeng [2,†], Fidoline Ngo Nonga [3], Daniel Frynta [4], Nwafi Ngeayi Adi [2], Jacob Willie [5,6] and Stijn Speelman [7]**

[1] African Wildlife Foundation, Yaounde P.O Box 5333, Cameroon
[2] Tropical Forest and Rural Development, University of Yaounde II, Yaounde P.O. Box 1365, Cameroon; tsafack.romaric@yahoo.com (R.T.D.); nwafi_adi2000@yahoo.com (N.N.A.)
[3] Faculty of Economics sciences, University of Yaounde II, Yaounde P.O. Box 1365, Cameroon; fiona_nonga@yahoo.fr
[4] Department of Zoology, Faculty of Science, Charles University, 38FQ+FX Prague, Czech Republic; frynta@centrum.cz
[5] Department of Biology, Faculty of Science, University of Ghent, 9000 Ghent, Belgium; jacob.willie@kmda.org
[6] Centre for Research and Conservation, Royal Zoological Society of Antwerp, 2000 Antwerp, Belgium
[7] Department of agricultural economics, Faculty of Bioscience Engineering, University of Ghent, 9000 Ghent, Belgium; stijn.speelman@ugent.be
\* Correspondence: mepanda@awf.org
† This work was part of the Master thesis of the second author Romaric Tsafack Donkeng. Master program in Yaounde 2 University, Cameroon.

**Abstract:** A large community of scientists has demonstrated that millions of people located in tropical zones derive a significant proportion of their livelihoods from the extraction of non-timber forest products (NTFPs). Despite these results, questions remain as to whether the valorisation of NTFPs can sustainably contribute to the improvement of the livelihood assets of the extractors. This study therefore evaluated the contribution of NTFP valorisation to the livelihood assets of local people around the northern periphery of the Dja Faunal Reserve (DFR), East Cameroon. To achieve this objective, data collected from 215 households in 32 villages were analyzed using factor analysis, Mann–Whitney U tests, and structural equation modelling. The results suggest that NTFP valorisation significantly contributes to the livelihood assets of local people at the periphery of the DFR. However, NTFP revenue was not significant in predicting their livelihood assets. Moreover, the local conservation management practices were not significant in predicting the livelihood assets in the long run. The results also revealed that individuals who received training and capacity building on good practices such as efficient collection techniques, effective drying techniques, and good conservation techniques earned better revenues and the impact on their livelihood was more significant than for those who did not. These results therefore recommend that the way forward for NTFP valorisation lies at the level of improving its quality and the market.

**Keywords:** livelihood; NTFP valorisation; revenue; conservation management practices; Dja Faunal reserve; Cameroon

---

## 1. Introduction

The importance attributed to forest resources and the contribution non-timber forest products (NTFPs) provide to the livelihoods of rural communities living in and around forest zones has increasingly been recognized over the last decades [1–3]. "Social forestry, community forestry, joint forest management, and conservation and development projects are examples of some of the concepts that were developed to put forward the importance given to forest resources" (see [4] and references therein). It is estimated that about one billion of the world's poorest communities depend on resources from forests for their livelihood [5–7]. NTFPs can be defined as all-natural products which can be obtained from the forest, wood lands or agroforestry other than wood [8]. These different products are used as food or as food additives [8,9]. The forests of the Congo Basin which constitute the second-largest forest block on the planet and represent one of the richest areas of the world in terms of biodiversity, provide a safety net to over 130 million people, many of whom depend directly or indirectly on NTFPs [8,10].

The commercialization of NTFPs was proposed in the 1990s to be one of the strategies for reconciling the conservation of biodiversity and development objectives in forest-based communities [11–16]. The different NTFPs commercialized can be of animal or plant origin. They include nuts, mushrooms, cork, rattan, essential oils, wild fruits, herbs, spices, aromatic and condiments plants (from plant origin), and honey, silk, game, and bees from animal origin. These products have a positive impact on the livelihoods of local people. Also, for their livelihoods to be sustainably improved, getting access to capital assets is important [17–19]. They include human capital (e.g., skills, knowledge, and good health), social capital (e.g., networks and membership of groups}, natural capital (e.g., land, forest, and biodiversity stocks), physical capital (e.g., shelter, water supply, and infrastructure), and financial capital (e.g., savings and money inflow) [20].

Despite the importance of NTFPs in the livelihoods and wellbeing of local people especially in the developing world, the sector still receives little attention in development policies and budgets, as well as in governments departmental budgeting and programming [21].

According to Ella Ella [22] "*the valorisation of NTFPs can not only divert local people from the illegal commercialization of bushmeat but can also contribute to ameliorating their income and their livelihoods, thereby reducing poverty*". NTFP valorisation means giving more value to NTFPs. This valuing can be economic in respect to the revenue and employment they provide to the extractors and others, social/cultural in respect to their different utilities (drugs, food, beverages etc.), or ecological in terms of the different eco-systemic services they offer [22].

Also, as emphasized by Shackleton [23], "*rather than assuming a universal positive relationship between NTFP use, forest conservation, and local livelihood, a much more location- and product-specific approach is needed. That is an approach where attention is not only given to the ecological characteristics of specific NTFPs, but also, to the nature of NTFP management practices and its value chains*". Within this framework, we carried out this research with the aim of evaluating the contribution of NTFP valorisation to the livelihood assets of local people at the northern periphery of the Dja Faunal Reserve (DFR). Specifically, we aimed to (i) evaluate the contribution of NTFP valorisation to the livelihood assets, (ii) evaluate the contribution of the revenue obtained from the commercialization of NTFPs to the livelihoods assets, and (iii) find out if conservation management practices used by local communities contribute to their livelihood assets.

## 2. Materials and Methods

### 2.1. Study Area

The study was conducted at the northern periphery of the Dja Faunal Reserve (DFR) (Figure 1) in the Messamena sub-division. The DFR is in southeast Cameroon with 4/5 of its area located in the Upper-Nyong Division (East) and 1/5 in the "Dja et Lobo" Division. The DFR is located between 2° 40′ and 3° 23′ N and 12° 25′ and 13° 35′ E, and its administrative area is 526,004 ha [24]. Three-quarters

of the perimeter of the DFR is delimited to the north, south, and west by the river Dja from which the reserve derived its name. The climate is of the humid equatorial type with four seasons—a long, wet season from August to November, a long, dry season from November to March, a short, wet season from March to June, and a short, dry season from June to August [25]. The average annual rainfall is 1563 mm and the average temperature varies between 19.8 °C and 27 °C [26]. Many ethnic groups, including the Badjoué, the Fang, the Kaka, the Nzime, the Boulou, the Njem, and the Baka, inhabit the periphery of the reserve.

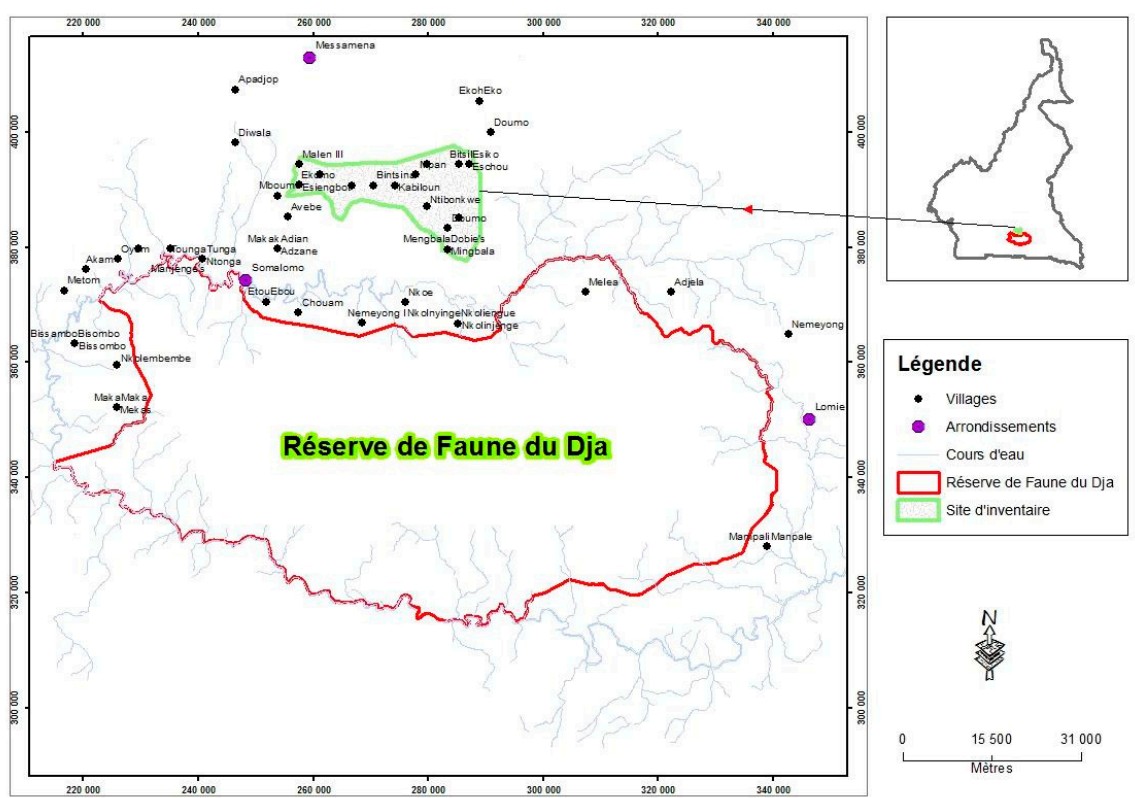

**Figure 1.** Location of the study area at the northern periphery of the Dja Faunal Reserve.

According to the 2005 Cameroon population census, the Messamena sub-division had a population size of 26,153 inhabitants made up of 13,441 males and 12,712 females [27]. The population density is not high—about 1.5 inhabitants/square km [27]. Most of the population depends on farming as their livelihood activity (about 80% of the population) [28].

Since 2010, Tropical Forest and Rural Development (TF-RD) a local non-governmental organisation (NGO) intervening in and around protected areas in Cameroon has been working with the local communities to create a socio-economic environment that supports empowerment of local communities and contributes to the conservation of biodiversity. This work has been done mainly by raising awareness on the importance of biodiversity conservation, environmental education, and the creation of alternative income-generating activities. To promote the sustainable extraction of NTFPs and to increase extractor's revenue, TF-RD developed an innovative approach to valuing these products based on training and capacity-building of the extractors on good harvesting methods, good cleaning and drying methods, good conservation methods, and the planting of new trees to sustain the specific species (first transformation process). Extractors that adopted this approach also received technical support for promotion and access to high value markets. Two types of extractors where identified around the research area—those who received training and adopted the innovative approach of extracting NTFPs and those who voluntarily maintained the local extractive methods (that is, cutting down trees

to collect the products, not drying the products before sale, and not planting new trees to sustain the species). The presence of these two groups gave us the opportunity to test the impact of the innovative approach.

*2.2. Data Collection*

A questionnaire administered from November 2016 to February 2017 was used as primary data collection instrument combined with the personal observation of the researchers. Data were collected using a semi-structured questionnaire. The questionnaire design was greatly facilitated by using survey instruments that pursued similar objectives and that had been adapted, tested, and effectively used to generate data in similar economic, agricultural, cultural, and ecological settings. Interviews were conducted in French because it is spoken by most of the population after their local language (Badjoue). However, to avoid any problems of interpretation, the interviewer was accompanied by a native field agent who served as a translator when necessary. Respondents were asked about household demographics, education, income-generating activities, different NTFPs extracted, places of extraction, and the modes of commercialization. The latent variables of NTFP valorisation, NTFP revenue, conservation management practices, and livelihood assets were measured using 27 items on a 6-point Likert scale (for more information on latent variable construction, see the data analysis section) with semantic differential statements (strongly disagree, disagree, moderately disagree, moderately agree, agree, and strongly agree). The questionnaire was pre-tested on a sample of 15 respondents in the villages of Ntoumzok, Nemeyong, and Kabilone II. Because of pre-testing and discussion, some questions were improved, and others deleted in order to improve clarity.

*2.3. Sampling Techniques and Procedure*

Out of the 81 villages that form the Messamena council, 32 villages were selected. These villages were selected because they are among the villages that benefited from a community-centered conservation program carried out by TF-RD. The population was divided into clusters of villages, and a specific sampling approach was applied in selecting the respondents within the sample. Using the information from the 2005 Cameroon General Population and Habitat Census (RGPH), the population of the study area was estimated to be 6227 inhabitants. To estimate the current population this number was updated assuming an annual population growth rate of 1.7% between 2005 and 2011 and 0.9% from 2011 to 2017. As a result, the population size at the time of the study was estimated to be about 7239 inhabitants. Since the study focused on households, the total number of households in the study area was estimated to be 1316 households. This number was obtained by dividing the total population by the average household size, which are 5.5 in rural areas in Cameroon [29]. The sample size of this study was 215 households accounting for 16.34% of the total number of households in the study area. A household was considered as being a group of individuals living in the same house and using the same pot. The study participants were selected using convenience sampling which is a non-probability sampling technique where subjects are selected because of their accessibility and convenience within the context of the research. The sample villages were Bintsina, Bitsil, Doumo Mama, Doumo Pierre, Echou, Kabilone II, Kompia, Madjuih II, Malen II, Malen V, Medjoh, Mimpalla, Nemeyong, Ngouleminanga, Ntibenkeuh, Lamakara, Djassa, Pallisco, Kabilone I, Malen III, Eko'o, Mpan, Bellay, Nkonzuh, Madjuih I, Palestine, Bifollone, Ntoumzok, Njolé Mpoum, Nkoul, Mboumo, and Bibom. The individual to whom the questionnaire was administered had to be a member of the sampled villages; he/she should be extracting NTFPs and should be present in the village during the sampling process.

*2.4. Method of Data Analysis*

This study used descriptive statistics and the partial least square structural equation modelling (PLS-SEM), otherwise known as soft modelling to present the results of the work. This method (PLS-SEM) was preferred to other testing methods such as the multinomial test and other correlation

tests (Pearson, Kendal, or Spearman) because it has several advantages (e.g., a normal distribution of the data is not assumed). This means that data with a non-normal distribution can be used when conducting structural equation modelling. In addition, indicators (items) with fewer than three occurrences for each construct could be used since the identification issue (variables) has been overcome. Furthermore, this data analysis approach can include many indicator variables (even over 50 items). It is used when dealing with weak theory, and it is a robust and flexible modelling approach [30]. This method was accompanied by a covariance-based approach of structural equation modelling (CB–SEM), which is well known for its accurate measurement of goodness of fit of indices. It is recommended to use at least three fit indices by including one index from each category of model fit [31].

Likert scales were used to measure how much respondents agreed or disagreed to the questions that were formulated. The 27 Likert scale items (Appendix A) in the questionnaire were subjected to a principal component analysis (PCA) using the statistical package for the social sciences (SPSS) version 21. Prior to performing the PCA, the suitability of the data for a factor analysis was assessed. The inspection of the correlation matrix revealed the presence of many Cronbach alpha coefficients of 0.3 and above. The Kaiser–Meyer–Olkin measure of sampling adequacy value was 0.807, exceeding the recommended value of 0.6; the Bartlett's test of sphericity was statistically significant ($p < 0.001$), thus supporting the factorability of the correlation matrix. This result indicated that the correlation structure was strong enough to conduct a factor analysis on the items. It therefore indicated that there was no issue of multi-collinearity in the data. In addition, the factor analysis suggested a linear relationship between the items.

For more convenience, a principal component analysis was carried out.

The result in Table 1 shows the various components extracted using principal component analysis. Principal component analysis aims at reducing a large set of variables to a small set that still contains most of the information in the large set. The technique of principal component analysis enabled us to create and use a reduced set of variables. The resultant principal component analysis revealed the presence of six components with eigenvalue values exceeding 1, explaining 14.36%, 11.67%, 9.96%, 9.65%, 9.05%, and 7.79% of the variance, respectively (Appendix B).

**Table 1.** Rotated component matrix/principal component analysis.

| Items | Components | | | |
| | Livelihood Assets | NTFP Valorisation | Conservation Management Practices | NTFP Revenue |
| --- | --- | --- | --- | --- |
| Item8 | 0.767 | | | |
| Item3 | 0.717 | | | |
| Item6 | 0.633 | | | |
| Item2 | 0.613 | | | |
| Item4 | 0.605 | | | |
| Item25 | | 0.714 | | |
| Item16 | | 0.670 | | |
| Item14 | | 0.605 | | |
| Item18 | | 0.569 | | |
| Item24 | | | 0.799 | |
| Item15 | | | 0.673 | |
| Item9 | | | 0.564 | |
| Item13 | | | | 0.776 |
| Item10 | | | | 0.700 |

Extraction method: principal component analysis. Rotation method: matrix with Kaiser normalization.

These items were subjected to further investigation using confirmatory factor analysis (CFA) (Figure 2).

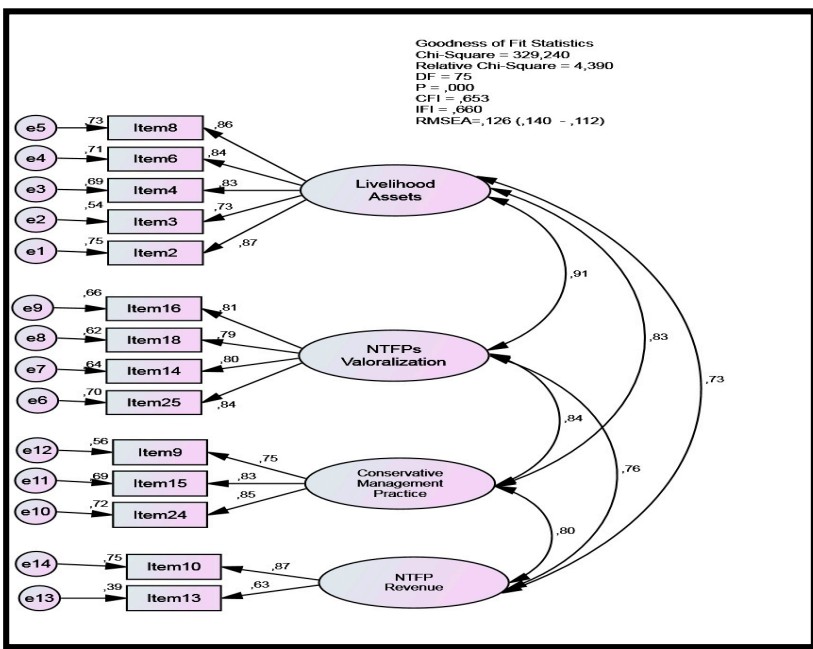

**Figure 2.** Confirmatory factor analysis (CFA) showing the regression between livelihood assets, non-timber forest product (NTFP) valorisation, conservative management practices, and NTFP revenue.

The CFA was conducted to measure the relationship between the observed and the underlying latent variables. In general, four fit indices were tested to determine the fitting of the model with the data. These were the chi-square statistic, the normed chi-square, the root means square error of approximation (RMSEA), and the comparative fit index (CFI). In these circumstances even if the model specification is present, it is recommended that for the model to fit the data, the value for the normed chi-square (CIMDF) should be less than 5, the root mean square error of approximation (RMSEA) should be less than 0.08, and CFI values are to be above 0.9. The result of the minimum value of discrepancy between the observed data and the hypothesized model divided by the degree of freedom (Cmin/df) in Figure 2 was significant (Cmin/df = 4.43). Incremental fit indices (CFI and NFI) did not surpass the threshold criteria value of being greater than 0.9 as recommended by Bentler and Bonett [32].

## 2.5. Validity of the Indicators

In the study, a convergent validity technique was used. This refers to the degree of agreement between two or more indicators of the same construct (dependent variables) [33,34]. Evidence of convergent validity was assessed by inspection of the loading factor, which indicates the weight of the relationship between the indicator and the construct. In other words, it is also referred to as the regression weight. For convergent validity to be established, the regression weight of the factor should be 0.5 or above. As observed in Figure 2, we can conclude that there is evidence of convergent validity between the indicators and the respective constructs. The construct "Livelihood Assets" had five indicators (items 2, 3, 4, 6, and 8), while "NTFP Valorisation" had four indicators present (items 14, 16, 18, and 25) (Appendix A). "Conservation Management Practices", which were considered as the local measures that are taken by the extractors to reduce their negative impacts on the valued species (NTFPs), and which, in the present study, included measures such as good harvesting and extracting techniques, NTFP regeneration through the establishment of nurseries, NTFP extraction authorizations, and simple management plans of community forests had three items. These three items were items 9, 15, and 24. Finally, "NTFP Revenue" had two (items 10 and 13). The validity was further assessed by testing for the significance of the item construct relationship (Table 2) using the maximum likelihood estimates approach [35,36].

**Table 2.** Test of validity of the indicators (regression weights).

|  |  |  | Estimate | S.E. | C.R. | P | Label |
|---|---|---|---|---|---|---|---|
| Item2 | <— | Livelihood Assets | 1000 |  |  |  |  |
| Item3 | <— | Livelihood Assets | 0.521 | 0.037 | 14.052 | *** | par_1 |
| Item4 | <— | Livelihood Assets | 0.842 | 0.046 | 18.121 | *** | par_2 |
| Item6 | <— | Livelihood Assets | 0.832 | 0.044 | 18.799 | *** | par_3 |
| Item8 | <— | Livelihood Assets | 0.918 | 0.047 | 19.535 | *** | par_4 |
| Item25 | <— | NTFP Valorisation | 1.000 |  |  |  |  |
| Item14 | <— | NTFP Valorisation | 0.811 | 0.049 | 16.584 | *** | par_5 |
| Item18 | <— | NTFP Valorisation | 0.773 | 0.048 | 15.967 | *** | par_6 |
| Item16 | <— | NTFP Valorisation | 1.001 | 0.059 | 17.060 | *** | par_7 |
| Item24 | <— | Conservative Management Practice | 1.000 |  |  |  |  |
| Item15 | <— | Conservative Management Practice | 0.945 | 0.055 | 17.100 | *** | par_9 |
| Item9 | <— | Conservative Management Practice | 0.778 | 0.056 | 14.009 | *** | par_10 |
| Item13 | <— | NTFP Revenue | 0.516 | 0.053 | 9.761 | *** | par_11 |
| Item10 | <— | NTFP Revenue | 1.000 |  |  |  |  |

Where **S.E** = Standard Error; **C.R** = Critical Ratios and **P** = P-value/degree of significance (* = 10 %, ** = 5 % and *** = 1%).

Table 2 shows that when "Livelihood Assets" changed by 1, items 3, 4, 6, 8 changed by 0.521, 0.842, 0.832, and 0.918, respectively. These regression weight estimates, of 0.521, 0.842, 0.832, and 0.918, had a standard error of 0.037, 0.046, 0.044, and 0.047, respectively. Dividing the regression weight estimates by their respective standard errors gave Z-statistics, or critical ratios, of 14.052, 18.12, 18.79, and 19.53, for items 3, 4, 6, and 8, respectively. In other words, the regression weight estimates were 14.052, 18.12, 18.79, and 19.53 standard errors above zero, for items 3, 4, 6, and 8, respectively. The probability of getting a critical ratio as large as 14.05, 18.12, 18.79, and 19.53 in absolute value is less than 0.001. In other words, the regression weights for livelihood assets for items 3, 4, 6, and 8 were significantly different from zero at the 0.001 level (two-tailed). The same explanation holds for the NTFP valorisation and its indicators (items 14, 16, 18, and 25), "Conservation Management Practices" (items 9, 15, and 24), and "NTFP Revenue" (items 10 and 13). The regression weights of these constructs were significantly different from zero at the 0.001 level.

### 2.6. Reliability of the Methods Used

The questionnaire was subjected to the reliability test using the Cronbach alpha test. The Cronbach alpha coefficient is an important measure of the internal consistency of the score. The Cronbach alpha value can lie between negative infinity and 1 ($-\infty < \alpha < 1$). It measures the degree to which the items that make up a scale are consistent. It is an indication of the extent to which the items are measuring the same underlying construct. Cronbach alpha values are interpreted as follows: for a value above 0.8, reliability is considered good; for a value between 0.6 and 0.8, reliability is considered acceptable; for a value below 0.6, reliability is considered unacceptable. In this study, the Cronbach alpha value was consistently good (= 0.835).

### 3. Results

Table 3 describes the sociodemographic characteristics of the respondents. From the 215 households sampled, 95% of the interviewed individuals were female with just 5% male. This indicates that the extraction of NTFPs is mainly carried out by the females. Of these extractors, 31.1% were aged 46 or above. This is because those interviewed in a household were mostly family heads (females) who had been carrying the activity over a long period. The results also suggest that the level of education was very low—46.5% of the respondents had no education certificate. A similar share (46.5%) has just a

primary education certificate. This is because most of those carrying the NTFP activities are female and mostly above 40. In addition, a high portion of the population in the study area had no education.

**Table 3.** Socioeconomic characteristics of the respondents.

| Name of Variables | Value of Variable | % | Names of Variables | Value of Variable | % |
|---|---|---|---|---|---|
| **Gender** | | | **Main Income Generating Activity** | | |
| Female | 205 | 95 | Farming | 124 | 42.9 |
| Male | 10 | 5 | NTFP extraction | 47 | 16.3 |
| **Age** Less than 18 | 3 | 1 | Retailer | 17 | 5.9 |
| 18–25 | 29 | 10 | Hunting | 5 | 1.7 |
| 25–30 | 32 | 11.1 | Fishing | 4 | 1.4 |
| 31–35 | 19 | 6.6 | Civil servant | 2 | 0.7 |
| 36–40 | 24 | 8.3 | Teacher | 1 | 0.3 |
| 41–45 | 18 | 6.2 | **Monthly earned income (US Dollar)** | | |
| 46 and above | 90 | 31.1 | Less than 55 | 144 | 67 |
| **Level of Education** | | | 56–90 | 47 | 22 |
| Primary | 100 | 46.5 | 91–135 | 15 | 7 |
| Secondary | 15 | 7 | 136–180 | 6 | 3 |
| No formal education | 100 | 46.5 | 181–265 | 1 | 0.4 |
| | | | 266–355 | 1 | 0.3 |
| | | | 356–450 | 1 | 0.3 |

Agriculture was the main income generating activity (42.9%), followed by the extraction of NTFPs (16.3%). Most of the respondents (144 individuals) had a monthly income of less than USD 55 (67%) [37], while 47 of them had a monthly income varying of USD 91–135 (22%). Of the respondents, 86.5% indicated that they extracted NTFPs from the forest and from their agricultural land, 2.3% indicated that they only extract from their agricultural land, and 8.4% only from the forest. Six respondents collected NTFPs within the village.

A total of 135 respondents extracted NTFPs for both consumption and commercialization, 23 of them extracted for commercialization, 9 extracted to offer as a gift, 1 extracted for use as medicine, and 42 indicated that they extract NTFPs for commercialization, consumption, and medicine. The different NTFPs extracted (that is their fruits and seeds) by the local people comprised *Baillonella toxisperma*, *Ricinodendron heudoletti*, *Irvingia gabonesis*, *Pentaclethra macrophylla*, and *Allanblackia floribunda* (common names are *moabi*, *njansang, bush mango, mbalaka,* and *bié*, respectively). The extraction of these NTFPs was also related to the accessibility and distance of these products from the villages, as it was observed that the quantity extracted was strongly correlated with the distance between the village and the site of extraction. The results of the survey indicated that 86.5% of the volumes of NTFPs extracted by the local community were obtained from both the forest and in farms, while 2.3% was from the village surroundings, and 2.3% from the village farm only. In addition, the mean revenues that the entire population obtained from NTFPs sold in the market were USD 9.91 for *Baillonella toxisperma*, USD 49.78 for *Irgingia gabonesis*, USD 27.75 for *Ricinodendron heudoletti*, USD 21.56 for *Pentaclethra macrophylla*, and USD 0.22 for *Allanblackia floribunda*. It should be noted that this great variation in the mean revenues earned was largely influenced by the demand for these products in the markets as well as their annual production.

Furthermore, in order to find out if there was a difference between the average income earned by those who voluntarily decided to adopt the innovative approach of extracting NTFPs (those who received training and capacity building on good practices such as efficient collecting techniques, effective fermenting period, effective drying techniques, and good conservation management) and those that maintained the local methods of extraction (these were members of the same villages who decided not to change their traditional way of extracting products, that is, not selecting the products to extract, not following the optimal drying procedure and fermenting period, and not planting trees), a Mann–Whitney U test was conducted. The results, in Table 4, revealed that those who

adopted the innovative approach of extracting NTFPs earned more income (the average individual income of a group member was USD 120.14) than those who maintained the local/traditional method (their average income was USD 95.63). The associated Mann–Whitney test showed that there was a statistically-significant difference of 10% ($p = 0.102$) and the average difference between those who were trained and who adopted the new approach and those who did not was USD 24.52 (this was obtained by subtracting USD 120.15 from USD 95.64). This result also suggests that innovation adoption, promotion of quality products, and access to high value markets generate local skills, increase income, and improve perception toward conservation.

**Table 4.** Mann–Whitney tests.

| Null Hypothesis | Test | Sig. | Decision |
|---|---|---|---|
| There is no difference in the distribution of total revenue across the two groups (member and non-members groups) | Mann–Whitney U test | 0.102 | Reject the null hypothesis |

To provide answers to the research questions, structural equation modelling was performed. The fully-fledged structural equation model in Figure 3 depicts all the standardized path coefficients among the latent constructs of the hypothesized theoretical framework. It was observed that three path coefficients were statistically significant at $p < 0.001$, one was significant at $p < 0.05$, and the remaining three appeared to be non-significant as evidenced by their critical ratios. From Figure 3, the hypothesized path coefficients of the results related to our specific objectives were obtained as seen in Table 4.

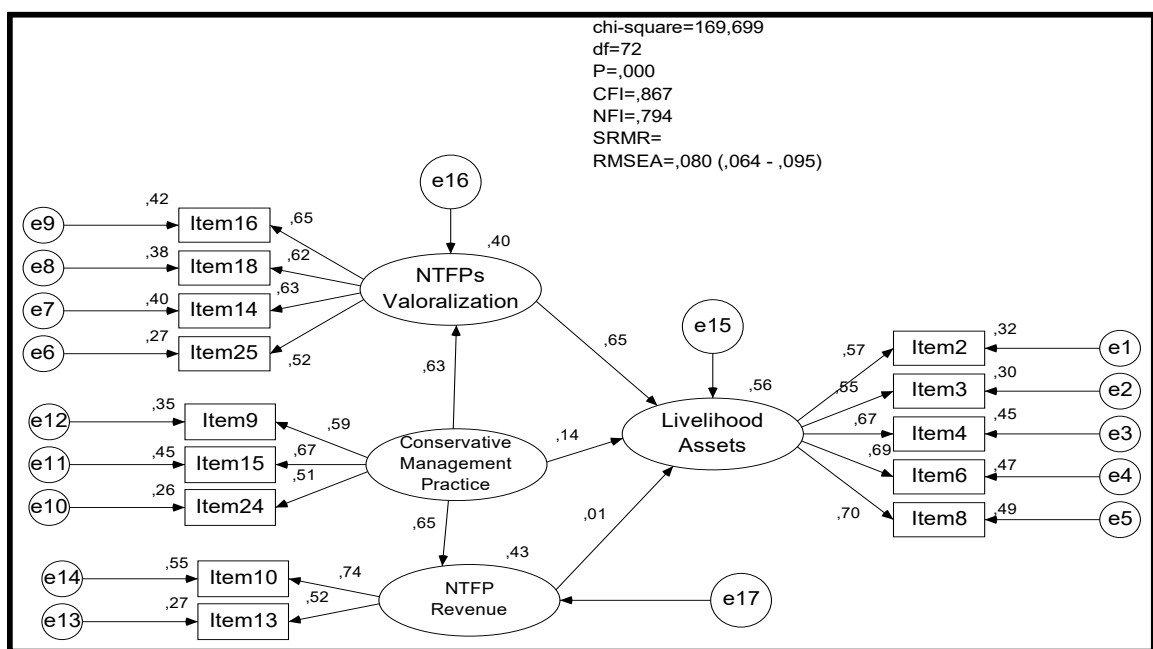

**Figure 3.** Results of the structural equation model.

This study shows that the valorisation of NTFPs significantly contributed to the livelihood assets of the local people around the northern periphery of the DFR ($p = 0.0001$). The study results indicate that NTFP valorisation yielded two major positive effects for the local people. Firstly, NTFP valorisation provided alternative income opportunities that enabled collectors to earn and save some of their income and secondly, the valorisation of NTFPs enabled the local people (mainly the women) to acquire a considerable supply of their household basic needs and food supplements.

Besides, the valorisation of NTFPs significantly contributed to the improvement of the human physical and financial capital assets (skills and knowledge for work and capacity building, access to information and sustainable management practices, and resulting income).

The income obtained from the commercialization of NTFPs, however, did not significantly predict the livelihood assets of the respondents ($p = 0.960$; Table 5).

**Table 5.** Hypothesized path coefficient.

| Exogenous Variable | | Endogenous Variable | Estimate | S.E. | C.R. | P | Decision |
|---|---|---|---|---|---|---|---|
| **NTFP valorisation** | <— | Conservative management Practice | 0.657 | 0.153 | 4299 | 0.0001 | Supportive |
| **NTFP revenue** | <— | Conservative management Practice | 1.126 | 0.236 | 4762 | 0.0001 | Supportive |
| **Livelihood assets** | <— | NTFP Valorisation | 0.639 | 0.161 | 3975 | 0.0001 | Supportive |
| **Livelihood assets** | <— | Conservation management Practice | 0.138 | 0.179 | 0.771 | 0.441 | Rejected |
| **Livelihood assets** | <— | NTFP Revenue | 0.004 | 0.080 | 0.050 | 0.960 | Rejected |

The results in Table 5 also indicate that actual conservation management practices used by those involved in the collection of NTFPs did not significantly predict the livelihood assets ($p = 0.441$). However, the test results further revealed that the conservation management practices used were significant in predicting the revenue obtained from the commercialization of NTFPs ($p < 0.0001$) and the valorisation of NTFPs ($p < 0.0001$).

## 4. Discussion

### 4.1. Contribution of NTFP Valorisation to the Livelihood Assets of the Local People

Similar results as those observed in Table 5 were obtained by Iponga and Isaiah Oino [38,39]. They also found that some of the reasons that could be at the heart of motivating local people to collect NTFPs were economic efficiency, social effectiveness, food production sufficiency, cash income, food security, and socio-cultural customs and obligations.

The results of this work also corroborate the findings of Rajib Biswal who studied the contribution of NTFPs to rural livelihoods in the Nilgiri Biosphere Reserve in India [40]. He also demonstrated that NTFPs significantly contribute to the livelihood assets of the rural communities around the Nilgiri Reserve. The contributions were classified according to the different components of livelihood assets, that is, the financial, human and physical capital assets. Moreover, our result also falls in line with the results obtained, which demonstrated that communities close to a reserve mostly rely on the reserve for firewood, medicinal herbs, fodder, and fruit nuts for household use and commercial sale.

### 4.2. Contribution of the Revenue Obtained from the Commercialization of NTFPs to the Livelihood Assets of Local People

The results obtained above were also observed by Cavendish [41] and Wunder [42] where they found that NTFPs mostly have a subsistence value, either for food, construction purposes, or household utensils, and that the extraction activity is a part-time activity for the people in rural areas, accounting for their subsistence needs and cash income. Also, some of the reasons that could explain this include the seasonality of the activity with the collecting periods usually starting in June and ending in November and late December according to the NTFP type, the low market prices and low demand for some of these NTFPs and the irregularity of the income obtained from their commercialization. These results were also confirmed by the FAO [43] and Browder [44].

In addition, other investigations indicate that NTFPs of plant origin currently represent the least important source of cash income and food for the livelihoods of rural people as compared to agricultural products and NTFPs of animal origin. A similar note of caution was also sounded by some other authors, indicating that the economic potential of most NTFPs is rather small [45,46] and that, from a monetary angle, NTFPs did not guarantee high or regular income for forest people [47,48].

### 4.3. Contribution of Conservation Management Practices to the People's Livelihood Assets

The effect of conservation management practices on livelihoods is consistent with the findings of Shiba [49] and Ticktin [50] who evaluated the harvesting practices of NTFPs from natural forests. They found that unsustainable NTFP harvesting practices are common, especially when leaves and barks are collected, and also that the population size and age structure of trees and plants were frequently affected by the exploitation (a decrease in the number of individuals of leaves, fruits, and seeds were noticed in at least one of the population age classes). Some of the reasons that could explain the patterns observed in this study include, for example, the vulnerability of these NTFPs to exploitation. For instance, *Baillonella toxisperma* (moabi) trees (which apart from being exploited for their fruits and bark are also exploited for their timber) are sensitive to exploitation because of their slow rate of re-barking (2 to 3 years), lack of knowledge concerning sustainable ways of extracting their products, poor harvesting techniques, and poor use of harvesting tools.

In addition, similar results were obtained by Snook [51], Ahenkan [52] and Arnold [53] stating that communities close to forest areas in Africa extract NTFPs from the forests irrespective of the management regimes or property rights in place and these are also related to the poor returns from agriculture and other off-farm income activities leading to more pressure exerted on forest resources and NTFPs.

Differences were also observed between the extractors that voluntarily decided to adopt the innovative approach and those that did not, concerning their behaviour towards conservation management practices. Differences were mainly related to their perceptions toward forest resources (mainly NTFPs), their understanding of sustainable development and sustainable harvesting and collection techniques, and the application of the simple management plan of their community forest.

## 5. Conclusions

This paper assessed the contribution of NTFP valorisation to the livelihood assets of local people around the northern periphery of the DFR. The results revealed that the valorisation of NTFPs significantly improved the livelihoods of the local people, but the revenues obtained from the commercialization of NTFPs did not significantly contribute to their livelihood assets. It was also observed that the conservation management practices used were not sufficiently significant to influence the livelihood assets of the local people. Moreover, the study also revealed that those who were trained and voluntarily adopted the new approach improved their livelihood assets and earned more income from the commercialization of NTFPs compared to those who did not. In addition, they had a better understanding of conservation activities and adopted good extracting techniques. These results highlight the need for support for forest-based communities by organizations that will help improve their perception of conservation and build their capacity in terms of efficient use of NTFP resources and sustainable management practices. In addition, these results also revealed that the way forward for improving the valorisation of NFTPs lies in ameliorating its quality and the market.

**Author Contributions:** Conceptualization, M.A.E.; data curation, M.A.E., R.T.D., and N.N.A.; formal analysis, M.A.E.; methodology, M.A.E., R.T.D., F.N.N., and D.F.; writing—original draft, M.A.E., R.T.D., F.N.N., and D.F.; writing—review and editing, M.A.E., R.T.D., J.W., and S.S. All authors have read and agreed to the published version of the manuscript.

**Funding:** This research was funded by MINRESI French Debt Relief Program (C2D/PAR).

**Acknowledgments:** We are very grateful to the team of Tropical Forest and Rural Development for their field assistance. We thank the people of the study area for their cooperation and active participation in this research and for the reliable information they provided during data collection.

**Conflicts of Interest:** The authors declare no conflict of interest.

## Appendix A

**Table A1.** Nature of the indicators (variables).

| Items | Nature of the Indicator |
|---|---|
| 1 | The sustainable management of the forest influences the quantities of the NTFPs collected. |
| 2 | The different uses of NTFPs have a significant impact on the life of the human population. |
| 3 | The income obtained from the commercialization of NTFPs ameliorates the living standards of the human population. |
| 4 | The income obtained from NTFP commercialization greatly ameliorates well-being. |
| 5 | The size of the household influences the quantity of NTFPs collected. |
| 6 | The distance of the village to the forest influences the quantity of NTFP collected |
| 7 | The economic status of the household (rich, poor, or moderate) influences the quantity of NTFPs collected. |
| 8 | The surface area cultivated influences the quantity of NTFPs collected. |
| 9 | The use of the simple management plan for community forest and NTFP extraction authorization can contribute in sustainably harvesting NTFPs. |
| 10 | The population density in the village influences the quantity of NTFPs collected. |
| 11 | The interval of time between NTFP collection and the other activities influences the quantity collected. |
| 12 | The marital status of the household influences the quantity of NTFPs collected. |
| 13 | The market price of NTFPs influences the quantity collected. |
| 14 | The perception of the forest by the population influences forest management. |
| 15 | The regeneration of NTFPs contributes to forest conservation. |
| 16 | The age of the household influences the quantity of NTFP collected. |
| 17 | The education level of the household influences the quality and quantity of NTFPs collected |
| 18 | Conservation education given by TF-RD influences the sustainability of NTFPs. |
| 19 | The extraction of NTFPs should not harm the plant. |
| 20 | Some NTFPs offers numerous uses (drugs, beverages etc.). |
| 21 | The lack of education on sustainable management influences the sustainability of NTFPs. |
| 22 | Different levels of capacity building support influence the quantity NTFPs collected |
| 23 | The consumption of NTFPs influences local peoples' food security. |
| 24 | The lack of good harvesting techniques and education on the simple management plan influences the harmful harvesting of NTFPs. |
| 25 | The transformation of NTFPs before their commercialization increases revenue. |
| 26 | The technique of collection of NTFPs influences the quantity produced by the plant. |
| 27 | The valorization of NTFPs can be an alternative to poaching. |

## Appendix B

**Table A2.** Total Variance Explained

| Component | Initial Eigenvalues | | | Extraction Sums of Squared Loadings | | | Rotation Sums of Squared Loadings | | |
|---|---|---|---|---|---|---|---|---|---|
| | Total | % of Variance | Cumulative % | Total | % of Variance | Cumulative % | Total | % of Variance | Cumulative % |
| 1 | 5105 | 26.868 | 26.868 | 5105 | 26.868 | 26.868 | 2730 | 14.368 | 14.368 |
| 2 | 1777 | 9352 | 36.220 | 1777 | 9352 | 36.220 | 2219 | 11.679 | 26.047 |
| 3 | 1574 | 8283 | 44.503 | 1574 | 8283 | 44.503 | 1893 | 9962 | 36.009 |
| 4 | 1267 | 6667 | 51.170 | 1267 | 6667 | 51.170 | 1835 | 9659 | 45.668 |
| 5 | 1124 | 5915 | 57.085 | 1124 | 5915 | 57.085 | 1720 | 9053 | 54.721 |
| 6 | 1033 | 5435 | 62.521 | 1033 | 5435 | 62.521 | 1482 | 7.799 | 62.521 |
| 7 | 0.789 | 4154 | 66.675 | | | | | | |
| 8 | 0.762 | 4012 | 70.687 | | | | | | |
| 9 | 0.700 | 3686 | 74.373 | | | | | | |
| 10 | 0.671 | 3530 | 77.903 | | | | | | |
| 11 | 0.633 | 3332 | 81.235 | | | | | | |
| 12 | 0.566 | 2981 | 84.216 | | | | | | |
| 13 | 0.555 | 2923 | 87.139 | | | | | | |
| 14 | 0.509 | 2680 | 89.819 | | | | | | |
| 15 | 0.498 | 2619 | 92.438 | | | | | | |
| 16 | 0.422 | 2220 | 94.658 | | | | | | |
| 17 | 0.415 | 2186 | 96.844 | | | | | | |
| 18 | 0.366 | 1925 | 98.769 | | | | | | |
| 19 | 0.234 | 1231 | 100.000 | | | | | | |

Extraction method: principal component analysis.

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
