# Peer review of "Contribution of Non-Timber Forest Product Valorisation to the Livelihood Assets of Local People in the Northern Periphery of the Dja Faunal Reserve, East Cameroon"

_forests, doi:10.3390/f11091019_

Round 1

Reviewer 1 Report

The research is interesing and the paper is well-written. Some more detailed information is missing, such as definitions of NTFP, NTFP valorisation, and the exact variables used in the statistical analyses. I would suggest also more detailed analysis of socio-ecomomic characteristics of the two groups of NTFP extractors (trained and non-trained) to look for the  differences that might affect incomes from selling these products. Some minor technical remarks are also in the text.

Author Response

Dear Reviewer,

Thanks for your comments. Here attache the file with answers to comments and observations.

The revised version of the article will be send to the editor.

Regards.

Reviewer 2 Report

The manuscript: “Contribution of non-timber forest products valorization on the livelihood assets of local people at the northern periphery of the Dja Faunal Reserve, East Cameroon”   deal with important  problem - nature protection and livelihood of local people. Material presented in the study is valuable but the method of calculation and presentation of results need substantial rearranging/improvement is not clear and is very difficult to understand.

Specific comments:

Line 39 (Yemuru et al., 2010) it should  be citing according to “Forests” standard [1], remark concern all manuscript.

Line 86 - Title under figure with capital letter at the beginning.

Line 88 References Tropical Forest and Rural Development (TF RD)1 should be cited according to Forests standard

Line 92 “… activities. in other “ why is full stop

Line 101 specie - it should be species

Line 103 - 107 This subparagraph is very short - two sentences - should better described how data was collected.

Line 109 - the abbreviation  “TF-RDT”  was defined in line 88 as (TF RD

Line 125-128 there are 32 villages, but in  line 109 is 31

Line 127 Lallisco,Kkabilone I  insert space after comma

Line 128 Njole mpoum - capital letter  “Mpoum”

Line 135 “correlation test models” they are not models but correlation tests or coefficients

Line 136 … is not assumed2 references should be cited according to Forests standard the same in line 145.

Line 142 (CB -SEM) -  without space

Line 149 explain abbreviation  SPSS

Line 153 p= 0.000 is not proper  concern all manuscript

Line 157-158 - which components?

Line 160 “Fig 2” should be Fig. 2

Line 161 - Figure 2  - not clear should be improoved

Line 187 “ in Fig 2 above - when giving number “in figure 2” “above” is not necessary -  concern in all manuscript

Line 192 - Table is not sufficiently described / explained should be improved.

Line 194 - what is the connection with table?

Line 195-206 not clear- should be improved.

Line 2017 - paragraph  Results should start with some texst with citihg the table.

Line 218  - rearranged variable  - gender - name of variable female - value/feature of fariable

Present income for example in thousand instead of so big numbers  

Line 229 “US Doller” the name of currency is Dollar - all manuscript

Line 260 Title of Table 3 is not proper - test of what?

Line 271 Table 4 should be improved.

Author Response

Dear Reviewer,

I grateful for your comments and observations on our article.

Here in the attachment the file with answers on your comments and observations.

I will send the revised full article to the editor.

Regards.

Round 2

Reviewer 2 Report

The second version of manuscript: “Contribution of non-timber forest products valorization on the livelihood assets of local people at the northern periphery of the Dja Faunal Reserve, East Cameroon” in my opinion has been significantly improved and can be published in Forests (all my remarks were considered) .

I still have one remark:

“Line 153 p= 0.000 is not proper concern all manuscript”. Line 153: Correction has been done, it is “p=0” all through the manuscript.

p-value can be very close to “0” (indicate high significance level) but not p=0.

I suggest use p < 0.001 or p < 0.0001

Author Response

Dear Reviewer,

Sorry for the late feedback. I have been in the field since Monday with few access to internet.

Please find in the attachment the answer to the comments and observations.

Regards 
